# Indole-Containing Metal Complexes and Their Medicinal Applications

**DOI:** 10.3390/molecules29020484

**Published:** 2024-01-18

**Authors:** Zahra Kazemi, Hadi Amiri Rudbari, Nakisa Moini, Fariborz Momenbeik, Federica Carnamucio, Nicola Micale

**Affiliations:** 1Department of Chemistry, University of Isfahan, Isfahan 81746-73441, Iran; f.momen@chem.ui.ac.ir; 2Department of Chemistry, Faculty of Physics and Chemistry, Alzahra University, Vanak, Tehran 19938-91176, Iran; n.moini@alzahra.ac.ir; 3Department of Chemical, Biological, Pharmaceutical and Environmental Sciences, University of Messina, Viale Ferdinando Stagno D’Alcontres 31, 98166 Messina, Italy; carnamuciof@unime.it

**Keywords:** indole, metal complexes, medicinal applications, biological activity

## Abstract

Indole is an important element of many natural and synthetic molecules with significant biological activity. Nonetheless, the co-presence of transitional metals in organic scaffold may represent an important factor in the development of effective medicinal agents. This review covers some of the latest and most relevant achievements in the biological and pharmacological activity of important indole-containing metal complexes in the area of drug discovery.

## 1. Introduction

Structurally, indole is a planar bicyclic molecule in which a benzene ring is fused to the 2,3-positions of a pyrrole ring (Figure 1). According to Huckel’s rule, indole is aromatic in nature [1]. In 1866, indole was synthesized for the first time by Adolf von Baeyer from the oxidation of indigo [2]. Since then, a variety of synthetic strategies have been developed for the preparation of the indole [3,4,5,6], including the Bischler-Möhlau [7], Fischer [8], Hemetsberger [9], and Julia synthesis [10]. In addition, various indole derivatives can be synthesized by substituting the indole ring at the N-1, C-2 to C-6, or C-7 positions with the ultimate goal of improving its characteristics [11].

The indole scaffold is known to be a key structural component of some classes of FDA-approved drugs such as the vinca alkaloids vinblastine and vincristine and the related semi-synthetic derivatives vindesine and vinorelbine (for the treatment of various types of cancer), physostigmine (for the treatment of Alzheimer’s disease and glaucoma), and alkaloids of the *Rauvolfia* species including ajmaline (1-A anti-arrhythmic agent), ajmalicine and reserpine (antihypertensive drugs) [12,13,14]. Furthermore, the presence of the pyrrole ring in the structure of indole makes the latter an electron-rich aromatic scaffold with distinctive binding properties, many of which occur with target-receptors belonging to the class of integral membrane G-protein coupled receptors (GPCRs) via complementary interactions with conserved binding pockets [15,16]. Like other heterocycles, indole is regarded as a main constituent of biomolecules and natural products endowed with outstanding properties. Tryptophan (Trp), an essential amino acid, is one example of them, whose structure possesses an indole group near the active site of metalloenzymes involved in electron-transfer pathways [17,18]. Other important examples include the phytohormone of the auxin class indole-3-acetic acid (IAA) and the metabolites/nutrients indole-3-carbinol (I3C) and 3,3′-diindolylmethane (DIM) which derive from the breakdown of the glucosinolate glucobrassicin present in high levels in vegetables of the *Brassicaceae* family (Figure 2).

Additionally, indole is regarded as a privileged molecular scaffold due to its presence in a wide range of pharmacologically active molecules and its ability to interact efficiently with multiple receptors besides GPCRs [19]. This wide range of biological activities of the indole scaffold includes antibacterial [20], anti-tubercular [21], α-amylase and monooxime inhibition [22], antiprotozoal [23], and anti-tumor activity [24]. Therefore, attempts to exploit this unique scaffold have attracted great attention from scientists. For instance, A. M. Bianucci et al. reported the key role of the indole N-H group in the interaction with specific receptors. Indeed, the isosteric replacement of the indole scaffold with a benzothiophene or benzofuran in a series of glyoxylylamine derivatives resulted in a decrease of the affinity for benzodiazepine receptors (Figure 3) [25].

The indole scaffold has also been successfully employed for the preparation of macrocyclic structures, which turned out to be effective as antimicrobial and anticancer agents [26,27,28,29,30,31,32]. Indole has been extensively studied in the context of the catalytic asymmetric synthesis for the development of indole-based chiral heterocycles, another important class of compounds that can be found in numerous pharmaceuticals, natural products, functional materials, chiral catalysts and ligands [33]. Nevertheless, indole derivatives have gained importance in the design and development of drugs to treat a variety of neurodegenerative disorders since the discovery of the indole derivative NC008-1 [34,35,36,37].

The pharmacological properties of these heterocyclic compounds can be considerably improved by increasing their solubility and bioavailability, which can be accomplished through the formation of complexes with a variety of transition metal cations [38,39]. Indeed, the coordination behavior of indole derivatives with transition metal ions has recently attracted the attention of researchers, and myriad publications using these compounds as ligands have been reported [40,41,42,43]. Koji Yamamoto et al. [44] reported different patterns of coordination of the indole rings with metal centers. They hold an opinion that pure σ-complexes are not always formed in indole–metal complexes; there is also the possibility of σ/π-intermediates (B) and π-modes (C or D) (Figure 1). Indole can exist in two tautomeric forms, 1*H*- and 3*H*-indole (Figure 4) [45]. 3*H*- and 1*H*-indole behave differently toward metal ions since (N(1)) in the former, which is considered as an imine nitrogen, has the coordinating ability, while the pyrrole moiety of 1*H*-indole is capable of forming salts with alkali metal ions by deprotonation. The different types of bonding of the indole ring with transition metal ions have been reported. For instance, a σ-bond between Pd(II) and the pyridine-like nitrogen of the 3*H*-indole tautomer was stated for the first time [46]. Apart from this, a number of noncovalent interactions of indole rings with other molecules has been observed via hydrogen bonding, π–π stacking and cation–π interactions [45].

Starting from these premises, we have focused this review article on the recent developments of indole-containing metal complexes, highlighting their structural features and above all their biological activity profile for possible medical applications.

## 2. Monodentate Ligands

### 2.1. N-Donor

Novel ligands, namely N-alkylindole-substituted 2-(pyrid-3-yl)-acrylonitriles (**L1a**–**L1e**; Figure 2), were recently used for the preparation of (p-cymene)Ru(II) piano-stool complexes (i.e., **1a**–**1e**; Figure 2) [47]. The synthesis of these compounds was achieved through the replacement of the isovanillyl and veratryl moieties of known anticancer tyrphostin derivatives with the indole scaffold, with the aim of enhancing their biological activity profile. The in vitro cytotoxicity of the ligands and related complexes against HCT-116 colorectal carcinoma (both p53-wildtype and p53-knockout cells) and MCF-7 breast cancer cell lines was investigated by MTT assay. Among the ligands, **L1a** and **L1b** turned out to be the most potent derivatives against MCF-7 cells (IC_50_ in the low-micromolar range) and the two HCT-116 cell lines (IC_50_ in the sub-micromolar range), respectively. The anticancer activity of the complex **1a** was higher than its ligand and the reference drugs sorafenib, gefitinib, and NAMI-A. The authors also studied the interaction between the most active compounds **L1a**, **L1b** and **1a** with DNA using the ethidium bromide (EB) intercalation assay, demonstrating high binding affinity for the complex **1a** and no DNA binding for the ligands **L1a** and **L1b**. Additionally, several experiments were carried out on HCT-116 p53-knockout colon cancer cells to shed light on the possible multiple mechanisms of actions of these types of compounds. They included colony formation assays, induction of apoptosis (which occurred via caspase 3/7 activation), ROS formation (particularly higher for **L1b** and **1a** than for **L1a**), experiments with tumor spheroids and molecular docking studies. The latter, performed into the protein structures of EGFR and VEGFR-2, highlighted that the indole-based ligands **L1a** and **L1b** are potential protein kinase inhibitors with preferential activity against VEGFR-2. Overall, these studies underlined that indole scaffold was essential for the high antiproliferative activity [47].

### 2.2. S-Donor

In 2018, Ashiq Khan et al. [48] prepared and characterized Cu(I) and Ag(I) complexes by reaction of copper(I) or silver(I) halides with indole-3-thiosemicarbazone (Hintsc, **L2**) or 5-methoxy indole-3-thiosemicarbazone (5-MeOHintsc, **L4**) or 5-methoxy indole-N^1^-methyl-3-thiosemicarbazone (5-MeOHintsc-N^1^-Me, **L5**) (Figure 3). According to the obtained characterization results, the complexes showed a monomeric tetrahedral structure [MX(HL)(Ph_3_P)_2_] (in case of M = Cu, **L2**, X = I, **2**; Br, **3**; Cl, **4**; **L4**, X = I, **5**; Br, **6**; Cl, **7**; **L5**, X = I, **8**; Br, **9**; Cl, **10** and in case of M = Ag, **L2**, X = Cl, **14**; Br, **15**; **L3**, X = Cl, **16**, Br, **17**; **L4**, X = Cl, **18**, Br, **19**). Furthermore, the reaction of copper(I) halides with indole-N^1^-methyl-3-thiosemicarbazone (HIntsc-N^1^-Me, **L3**) resulted in the formation of dinuclear complexes; [Cu_2_(μ-X)_2_(η1-S-H^3^L)_2_(Ph_3_P)_2_] (X = I, **11**; Br, **12**; Cl, **13**). The characterization of **L3**, **4**, **8**, **9**, **11**, **12** and **14** was carried out by means of single-crystal X-ray diffraction (Figure 5), indicating that the metal ions in the complexes **4**, **8**, **9** and **14** were occupied by one halogen atom, two phosphorous atoms from two triphenylphosphine molecules and a thione sulfur atom from the thiosemicarbazone ligand. However, a halogen bridge connected two copper atoms in complexes **11** and **12**, forming a (Cu_2_(μ2-X)_2_Cu X = I, **11**; Br, **12**) core. A sulfur atom from the thiosemicarbazone ligand and a phosphorous atom of the triphenylphosphine molecule also coordinated to each copper atom in a trans-position. The authors reported the evaluation of the anti-M. tuberculosis activity of ligands (**L2**–**L5**) and their metal complexes (**2**–**19**) against M. tuberculosis H37RV strain ATCC 27294 in which they obtained significant outcomes (MIC = 1.6 µg/mL; much higher than first- and second-line drugs employed as positive controls) for the ligand **L4**, Cu(I) complexes **2** and **3**, and Ag(I) complex **14**. Furthermore, through docking studies the authors have identified the inhibition of enoyl reductase as a possible mechanism of these compounds [48].

## 3. Bidentate Ligands

### 3.1. O, O-Donor

In 2014, Asha Chilwal and co-workers synthesized six organotin(IV) complexes of Me_2_SnL_2_, Bu_2_SnL_2_, and Ph_3_SnL [where L = indole-3-butyric acid (IBH; i.e., **20**, **21** and **22** in Figure 6) or indole-3-propionic acid (IPH; i.e., **23**, **24** and **25** in Figure 6)] by reaction of the corresponding di-organotin(IV) oxide and triphenyltin(IV) hydroxide IBH or IPH in the desired molar ratios of 1:2/1:1 [49]. The characterization of all compounds was carried out by means of elemental analysis, thermogravimetry (TG) technique, IR, ^1^H NMR, ^13^C NMR, and ^119^Sn NMR spectroscopy. The structures of the complexes were suggested based on the magnitudes of the coupling constants 1J(^119^Sn–^13^C), namely a tetrahedral geometry around the tin atom in solution for the tri-organotin complexes (**22** and **25**) and an octahedral geometry in solution as well as in the solid state for the di-organotin(IV) complexes (**20**, **21**, **23**, and **24**) (Figure 6). In addition, the authors studied the effects of the metal complexes on three gram-positive (*S. aureus*, *S. epidermidis*, and *M. luteus*) and three gram-negative (*E. coli*, *P. aeruginosa*, and *E. aerogenes*) bacteria using the minimum inhibition concentration (MIC) method. Several important results were achieved: (1) the synthesized compounds were more active against gram-positive than gram-negative bacteria; (2) upon complexation, the antimicrobial activity of the compounds was remarkably enhanced realistically due to enhanced lipophilicity and permeation through the lipid layer of the bacterial membrane; (3) the synthesized compounds exhibited greater antibacterial activity as compared to the reference drug (chloramphenicol).

### 3.2. N, N-Donor

The reaction of the 2-imino-indole derivatives **L6**–**L11** (which were in turn synthetized by condensation of one equivalent of the appropriate aniline with one equivalent of 3-chloro-1H-indole-2-carboxaldehyde) with trans-chloro(1-naphtyl)bis(triphenylphosphine)nickel(II) [C_10_H_7_NiCl(PPh_3_)_2_] led to a series of neutral nickel complexes (i.e., **26**–**31**; Figure 4) [50]. The X-ray study of the complex **31** revealed that the nickel atom adopts a square-planar coordination geometry, wherein the 2,4,6-trimethylanil and naphthyl group occupy the position trans to the triphenylphosphine ligand (with a P1–Ni1–N2 angle of 174.56(12)°) and to N1 (with an N1–Ni1–C34 angle of 168.6(3)°), respectively (Figure 7). The catalytic activity of the complexes was investigated for ethylene oligomerization with no additional activator, displaying significant activity (up to 2.05 × 10^4^ g ethylene mol^−1^ h^−1^) for each of them according to following order: **30** > **28** > **27** > **29** > **31** > **26** [50].

Joan J. Soldevila-Barreda and co-workers proposed four half-sandwich complexes containing the 2-(2-pyridinyl)-1H-indole (ind-py) moiety as a ligand (i.e., **32**–**35**; Figure 8) [51], which were characterized by ^1^H and ^13^C-NMR spectroscopy and high-resolution ESI-MS. To evaluate their antiproliferative activity the authors performed the MTT assay by using two ovarian cancer cell lines (cisplatin sensitive A2780 and cisplatin resistant A2780cisR) and one normal prostate cell line (PNT2). This assay evidenced that the cytotoxicity of the complexes was overall lower than that of the reference drug cisplatin (range 13.0–62.8 µM). It is worth noting that the cytotoxic activity of the Rh(III) (**32**) and Os(II) (**35**) complexes were 2–3 folds higher towards A2780 with respect to normal cells. In addition to this, the authors investigated the out-of-cells catalytic activity of the compounds for reduction/oxidation of nicotinamide adenine dinucleotide coenzymes (NAD) through NMR spectroscopy. Based on the results obtained, the reduction of NAD+ occurred with all complexes and in the presence of sodium formate with turnover frequencies comparable to those of previously reported catalytic metallodrug candidates. The out-of-cell investigations included reaction with glutathione (100% of adduct formation) and with nucleobases (K_b_ in the range 10^3^–10^4^ M^−1^; then suggesting them as weak DNA binders). They also employed the co-incubation method with sodium formate and N-acetyl cysteine to study the in-cell catalytic activity of the complexes, concluding that only the Ir(III) (**33**) and Rh(III) (**32**) complexes are able to generate oxidative stress [51].

Six Zn(II) complexes (i.e., **36**–**41**; Figure 5) with the formula [Zn(InR-Im)_2_Cl_2_], where InIm is 3-((1H-imidazol-1-yl)methyl)-1H-indole, were designed and synthesized from their corresponding ligands (**L7**–**L12**; Figure 5) by K. Babijczuk and co-workers [52]. The investigation of their structures was accomplished through NMR, FT–IR and ESI–MS spectrometry, elemental analysis and single-crystal X-ray diffraction. According to the data obtained, complexes **36**, **37**, **38**, **39** and **40** are composed of a zinc ion coordinated by two imidazole nitrogen atoms of two indole–imidazole hybrid ligands and two chloride ions in a distorted tetrahedral environment. The crystal structure of complex **39** is shown in Figure 9 as an example of the determined similar structures. The comparison between the structure of the complex **38** and its uncoordinated ligand **L9** revealed that the absolute values of the torsion angles along the C–C (φ1) and C–N (φ2) bonds formed by the methylene bridge increases and decreases upon complexation, whereas these conformational changes are not significant in the case of the complex **40**. Hemolytic activity assays showed that only the complexes with electron-withdrawing groups in the imidazole ring (i.e., **40** and **41**) are notably cytotoxic (>5%) as compared to the free ligands. On the contrary, the complexes containing either an unsubstituted or electron-donor-substituted ligand at the same nucleus showed no toxicity. The cytoprotective activity of the complexes against AAPH-induced hemolysis was also studied, indicating that this activity increases upon complexation with ZnCl_2_ following the order **36** > **39** > **37** > **40** > **41** > **38**. Furthermore, these complexes turned out to be effective as antibacterial (in particular **36** against M. luteus; growth inhibition zone = 10.6 mm) and antifungal (in particular **41** against fungi of the genus Trichoderma) agents [52].

A series of luminescent rhenium(I) diimine–indole complexes (i.e., **42a**–**45b**; Figure 10) and their indole-free counterparts (**42c**–**45c**; Figure 10) were obtained by K. Kam-Wing Lo et al. in 2005 by using py-3-CONHC_2_H_4_-indole, py-3-CONHC_5_H_10_CONHC_2_H_4_-indole, or py-3-CONH-Et ligands as N donor in combination with diamine ligands, i.e., Me_4_-phen, phen, Me_2_-phen and Ph_2_-phen [53]. According to X-ray analysis, in the crystal structure of **44a** the Re(I) center adopted a distorted octahedral geometry and coordinated with two carbonyl groups in a facial orientation, while a dihedral angle of ca. 7.29° was formed from the indole unit and the Me_2_-phen ligand of the same molecule (Figure 11). In this complex, no stacking interactions were observed between the diimine ligand and the indole moiety of adjacent molecules. All newly synthesized complexes showed green to orange-yellow luminescence upon visible-light excitation. The indole-containing complexes’ spectra recorded in the ultraviolet region produced an additional emission band due to the indole moiety. The lower luminescence intensity of the complex compared to free indole is probably due to strong absorbance of rhenium(I)-diimine units at the excitation wavelength or resonance energy transfer from the indole to the luminophore. To gain insight into the quenching mechanism, Stern-Volmer studies on the indole-free complexes in the presence of indole as a quencher were performed. These additional studies confirmed the self-quenching of the indole-containing complexes, which stem from the intermolecular electron transfer. Moreover, the emission titration technique was employed to evaluate the bovine serum albumin (BSA)-binding of the diamine–indole complexes. This binding study clearly indicated that the indole moiety is responsible for the protein-complex formation, since no binding to BSA was observed in the indole-free complexes. This set of rhenium(I) diamine–indole complexes was also investigated for their ability to inhibit the bacterial enzyme tryptophanase (TPase). Also in this case, the binding of the complexes to the enzymatic protein (and the consequent inactivation) occurs due to the presence of the indole moiety [53].

Synthesis and study of the anticancer activity of novel indole-fused latonduine derivatives and their Ru^II^ and Os^II^ complexes (i.e., **L13**–**L14** and **46**–**49**, respectively; Figure 12) were reported by Christopher Wittmann et al. in 2022 [54]. All complexes were found to be chiral at the metal center and as a racemic mixture when in solution. The single-crystal structure of **46** was determined by X-ray diffraction method and revealed that the complex adopted the three-leg piano-stool geometry, in which the ruthenium(II) was coordinated to the two N-atoms of the ligand **L13**, one chloride ion and a p-cymene group (Figure 13). Furthermore, the obtained data displayed that the complex crystallizes as a racemate in the orthorhombic space group Pna21 with three-leg piano-stool geometry wherein the bidentate ligand **L13** and a chloride ion act as legs while the p-cymene group was as the piano stool. Both ligands and complexes were evaluated for their anticancer activity against MDA-MB-231, LM3 and U-87 MG cell lines. The Ru(II) and Os(II) complexes showed a lower efficacy (IC_50_ = 57–250 µM) compared to indole-based ligands (IC_50_ = 1.4–10 µM) and the reference drugs cisplatin, sorafenib and paclitaxel. MDA-MB231 cells were also employed for the assessment (fluorescence imaging technique) of the microtubule-destabilizing properties of these derivatives. This assay evidenced excellent tubulin-targeting effects for both ligands and metal complexes, in particular for the ligand **L13**, which proved to be even more potent and less toxic than the reference microtubule-destabilizing agent colchicine. The antitumor effect was also studied in vivo using two different mouse tumor models, namely MDA-MB-468 (breast cancer) and LX22 (small cell lung). Breast cancer-bearing mice developed ulcerative tumors rapidly after receiving a single injection of complex 1 (10 mg/kg, intravenously). These in vivo pilot experiments revealed also that complex **46** was well tolerated in mice; however, tumor growth was not significantly inhibited. Mice-bearing human LX22 lung cancer treated with complex **46** showed a 1.3-fold lower tumor volume in comparison to those of the control group [54].

A comprehensive study on metal complexes that have an indole derivative as a bidentate ligand has been conducted by Yareeb J. Sahar et al. [55]. By using (E)-2-(1H-indol-3yl)diazenyl)thiazole (HIDAT) ligand (i.e., **L15**; Figure 14) as a starting material, four novel complexes (cobalt, nickel, copper and palladium derivatives) (i.e., **50**–**53**; Figure 14) were obtained and characterized by means of different techniques. Based on the obtained data, the metal ions in the complexes exhibited an octahedral geometry except for Pd(II), which was square-planar. The Pd(II) complex **50** showed significant anticancer activity against the human leukemia cell line HL-60 (IC_50_ = 27.02 μg/mL) and moderate tumor selectivity evaluated towards HdFn healthy cells (IC_50_ = 83.69 μg/mL). Docking studies were performed to evaluate the interaction between this Pd(II) complex and the tyrosine-protein kinase ABL1 receptor, which is related to the emergence of leukemia according to recent studies [56]. In Figure 15, the main interactions of **50** with the target are displayed in 2D. The data obtained from these computational studies pertaining to the Pd(II) complex and the activity-related reference antineoplastic drug Nelarabine indicated that **50** possesses promising anti-leukemic activity. The high propensity of the complex to bind to the receptor may be due to the presence of the nitrogen and sulfur atoms of the heterocyclic indole and thiazole rings, respectively, in the structure of the complex [55].

The 3-methoxy-indole-hydrazone glyoxime ligand **L16** (Figure 16) was used for the preparation of the related Ni(II), Cu(II), and Co(II) complexes (**54**–**56**; Figure 16) and also its BF_2_^+^-bridged transition metal complexes (**57**–**59**; Figure 16) by Babahan I. et al. [57]. The latter were synthesized through the addition of boron trifluoride etherate complex to the solution of [M(L)_2_]. The Co(II) complex **56** showed an octahedral geometry with water molecules as axial ligands while a square-planar environment was suggested for the Ni(II) and Cu(II) complexes (**54** and **55**, respectively), highlighting the effect of metal ions on the complexes’ structures. Based on the spectral studies, the ligand acted as a neutral bidentate N, O-donor through the azomethine nitrogen atom and the imine oxime group. The antitumor potential of these complexes was evaluated against MCF-7 and PC-3 cell lines. Additionally, the Hoechst/propidium iodide double-staining method was employed to determine their apoptotic or necrotic effects towards cells. From these biological assessments emerged that all compounds were effective against both tumor cell lines in the range of 5–40 μM, suggesting apoptotic mechanisms. More importantly, it turned out that their cytotoxic activity was higher than that of the already approved anticancer drug paclitaxel used as a positive control [57].

### 3.3. N, O-Donor

Novel Ni(II) (**60** and **62**; Figure 17) and Cu(II) complexes (**61** and **63**; Figure 17) were prepared from the Schiff base ligand **L17** ((E)–2–(((5H–[1,2,4]triazino [5,6–b]indol–3–yl)imino)methyl)phenol), which was in turn synthesized by condensation of 5H-[1,2,4]triazino [5,6-b]indol-3-amine with a salicylaldehyde unit [58]. The Ni(II) center in **60** and **62** adopted a square-planar geometry, unlike **61** and **63** in which an octahedral geometry was observed around the Cu(II) center. Electronic absorption titrations and fluorescence spectral studies displayed an interaction of the complexes with DNA, probably by electrostatic surface binding mode along with partial intercalation in the minor groove mode. K_b_ values evidenced that the complexes containing phen **62** (1.9 × 10^4^ M^−1^) and **63** (4.8 × 10^4^ M^−1^) possess a greater CT-DNA binding capacity than the complexes with bpy **60** and **61**. The two phen derivatives showed also superior HSA-binding capacity compared to the bpy derivatives [58].

Two Co(II) and Zn(II) complexes containing an indole ring (i.e., **64** and **65**; Figure 18) with a bidentate ligand were proposed by Youssef Ghufran Shakir et al. [59]. The reaction of the combined ligand (i.e., **L18**; Figure 18), which was in turn synthetized by condensation of the diazonium salt of 4-aminoantipyrine and indole in basic conditions, with CoCl_2_·6H_2_O and ZnCl_2_ led to these complexes. ^1^H-NMR, IR, mass spectrometry, UV-Vis, powdered XRD, molar conductivity and magnetic susceptibility technique were utilized to determine the structure and properties of the ligand and its complexes. The cytostatic activity of the ligand and its complexes was investigated against MCF-7 tumor cell line and HdFn healthy cell line. All the examined compounds displayed higher cytotoxicity against MCF-7 cells than HdFn cells. Particularly, **L18** was more effective against MCF-7 cells as compared to the zinc complex **65**. Additionally, the well diffusion method was employed to investigate the antimicrobial activity of the compounds against *S. aureus* and *E. coli* bacteria, revealing superior inhibitory properties of the complexes compared to the ligand [59].

R. Reshma et al. [60] developed a series of Mn/Fe/Co/Ni/Cu/Zn(II)-(indal-L-his) complexes starting from the ligand **L19** (i.e., **66**–**71**; Figure 19). The indole-based ligand was obtained in turn by reaction of the indole-3-carboxaldehyde (indal) with L-histidine (L-his). The structure of all complexes was determined using elemental analysis, molar conductivity, magnetic, IR, UV–Vis, ^1^H NMR, mass and ESR spectroscopy, powder XRD and thermal gravimetric analysis (TGA). Interestingly, Mn(II) and Fe(II) complexes (**66** and **67**, respectively) adopted an octahedral geometry, [M(II)-(indal-L-his)_2_(H_2_O)_2_], whilst Co(II) and Zn(II) complexes (**68** and **71**, respectively) with [M(II)-(indal-L-his)_2_] formula possessed a tetrahedral geometry. A square-planar environment was observed for Ni(II) and Cu(II) complexes (**69** and **70**, respectively), [M(II)-(indal-L-his)_2_]. Based on the results of the antimicrobial tests (disc diffusion technique), all metal complexes exhibited higher activity than the indal-L-his ligand **L19** against Gram-positive bacteria (*B. subtilis* and *S. aureus*), Gram-negative bacteria (*P. aeruginosa* and *E. coli*) and fungal species (*A. niger* and *C. albicans*). The copper derivative **70** ([Cu(II)-(indal-L-his)_2_]) displayed the most significant activity amongst all complexes. The antimicrobial activity of this copper complex was superior to that of the antibacterial reference drug ciprofloxacin and antifungal drug fluconazole [60].

Synthesis and characterization of a series of Cu(II), Co(II), Ni(II) and Zn(II) complexes (i.e., **72**–**79**; Figure 20) containing amino acid-derived Schiff base ligands with general formula [ML_2_] (**72**–**75**) and [ML(1,10-phen)_2_]Cl (**76**–**79**) were achieved by Arunadevi A. and Raman N. [61]. The preparation of the ligand (**L20**) was carried out by condensation of the 4-chloro-3-nitrobenzaldehyde with 2-amino-3-(1H-indol-3-yl)propanoic acid. This series of complexes were assessed for their antimicrobial (antibacterial and antifungal) activity and binding properties towards biological targets (DNA and BSA). In detail, their binding to DNA was assessed by different physicochemical techniques (electronic absorption titration, fluorescence spectroscopy and viscosity measurements), which indicated that it occurs via intercalative mode. All complexes also showed high DNA cleavage capacity in the presence of activator (H_2_O_2_), unlike the parent ligand that did not cleave DNA under the same physiological conditions. As compared to all complexes, the mixed-ligand [ML(1,10-phen)_2_]Cl type **76** (Cu-derivative) and **78** (Ni-derivative) displayed the highest cleavage activity in accordance with their high DNA-binding affinity and structural features (presence of the auxiliary phen ligand that favor π–π stacking interaction with DNA base pairs). These two complexes were able to convert form I DNA into form II and even form III to a large extent, as assessed by gel electrophoresis. Ligand and both types of complexes were also tested against a selected panel of bacteria (*B. subtilis*, *S. aureus*, *P. aeruginosa*, *K. pneumonia* and *S. typhi*) and fungi (*A. niger*, *A. flavus*, *C. lunata*, *R. babaticola* and *C. albicans*). Overall, these studies highlighted the higher efficacy of the metal complexes compared to free ligand [61]. **L20** and the whole panel of metal complexes underwent also exhaustive computational studies which highlighted their drug-likeness profile for oral administration and corroborated the binding with DNA [62].

Synthesis and characterization of mononuclear rhenium(I) complexes (i.e., **80**–**84**, Figure 21) with bidentate indole-pyrazoline based ligands (i.e., **L21**–**L25**, Figure 21) derived from α,β unsaturated enons and benzhydrazide were reported by Reena R. Varma et al. [63]. The DNA-binding capacity, in vivo and in vitro cytotoxicity as well as the antimicrobial activity of the complexes and ligands were investigated. The results suggested DNA groove binding mode for all compounds with affinity in the order **84** > **80** > **83** > **81** > **82** > **L25** > **L21** > **L22** > **L23** > **L24**. All complexes also exhibited good antiproliferative activity against MCF-7, HCT 116, and A549 tumor cell lines. In particular, complex **84** showed the highest cytotoxicity (higher also than that of the anticancer drugs carboplatin and oxaliplatin used as positive controls). Using S. cerevisiae cells, the trend of the % residual cell viability followed a mixed trend between ligands and complexes, i.e., **84** > **82** > **L23** > **L25** > **81** > **80** > **L22** > **83** > **L21** = **L25**, and all compounds turned out to be more toxic than the positive controls employed in this assay (ciprofloxacin, norfloxacin and terbinafine). This yeast species was also used to demonstrate that these compounds are capable of eliciting oxidative stress via ROS production. The in vivo cytotoxicity was evaluated in terms of brine shrimp larvae death. Noteworthy, in this assay the ligands turned out to be more toxic than complexes with a mortality rate directly proportional to their concentration. However, also in this case, complex **84** was the most interesting derivative amongst the complexes and all compounds were less toxic than the reference drug cisplatin. The antibacterial assays (carried out on two Gram-positive and three Gram-negative bacteria) provided unequivocal MIC values in the order complexes > ligands > positive controls, confirming the importance of obtaining metal complexes [63].

Further advancements in the development of metal complexes endowed with indole rings in the structure of the ligand and containing manganese(II) as a transition metal were reported by S. Sharma et al. in 2016 [64]. In this research work, four bidentate ligands were synthesized, namely **L26** [2-(5-fluoro-2-dihydro-2-oxo-1H-indol-3-ylidene)hydrazinecarboxamide], **L27** [2-(5-fluoro-2-dihydro-2-oxo-1H-indol-3-ylidene)hydrazinecarbothioamide], **L28** [2-(5-bromo-2-dihydro-2-oxo-1H-indol-3-ylidene)hydrazinecarboxamide] and **L29** [2-(5-bromo-2-dihydro-2-oxo-1H-indol-3-ylidene)hydrazinecarbothioamide] (Figure 22). Their complexation with MnCl_2_·4H_2_O led to Mn(II) complexes (i.e., **85**–**92**; Figure 22) which, based on the spectral data, showed a tetrahedral geometry. Both the ligands and complexes showed acceptable toxicity against bacteria (*E. coli* and *S. aureus*) and fungi (*F. semitectum* and *A. flavus*), with the complexes being more active than the ligands. Coordination of metal ions correlates with enhancement of DNA-cleavage activity by the complexes, as demonstrated by gel electrophoresis experiments using DNA isolated from *E. coli* (ATCC 25922). Specifically, thiosemicarbazone complexes **86** and **90** exhibited better DNA-cleavage activity than the corresponding semicarbazone derivatives **85** and **89** [64].

Co(II) and Ni(II) complexes (i.e., **93**–**94**; Figure 23) were obtained by I. I. Seifullina and co-workers in 2020 via reaction of M(CH_3_COO)_2_ with 2-(7-bromo-2-oxo-5-phenyl-3H-1,4-benzodiazepin-1-yl)acetohydrazide (Hydr) and an indole derivative, i.e., 1H-indole-2,3-dione (HIz). The structure of the complexes was investigated using elemental analysis, thermogravimetry, IR spectroscopy, mass spectrometry and X-ray absorption spectroscopy. The results suggested an octahedral geometry surrounding the cobalt and nickel ions and composed of six oxygen and nitrogen atoms [65]. These complexes have not been subjected to biological evaluations but, considering their structural features (presence of the indole group plus the 1,4-benzodiazepine scaffold as potential β-turn mimetic), they deserve full attention in this regard.

The tridentate Schiff base ligands **L30**–**L33** (Figure 24), which were obtained by reaction of indole-3-butyric hydrazide with variously substituted salicylaldehydes, were used for the preparation of diorganotin (IV) complexes, R_2_SnL (i.e., **95**–**110**; Figure 24). The structure of the complexes was determined by using UV–Vis, FT-IR, NMR (^1^H, ^13^C, ^119^Sn), mass spectrometry and TGA, which showed that the metal ions of the dialkyl/diaryltin (IV) moieties were coordinated to two oxygen and one nitrogen atoms of the ligand in a pentacoordinated geometry. All compounds underwent antimicrobial screening against a selected panel of bacterial and fungal strains showing promising activity, with complex **110** (Phe_2_SnL33) as the most active derivative. They were also investigated as potential anticancer agents against two tumor cell lines (A549 and MCF-7) and one normal cell line (IMR 90). In this assay (MTT method), they showed moderate antiproliferative activity. The best compounds proved to be the ligand **L32** and the complex **100** (IC_50_ values in the micromolar range), which also exhibited a good selectivity index (~7–8) [66].

Another interesting class of indole Schiff base compounds were developed and fully characterized by F.L. Faraj and co-workers in 2014 [67]. The chelating ligand (obtained by condensation of 2-(diformylmethyl-idene)-3,3-dimethylindole with o-aminophenol) consisted of a meso-substituted β-diiminate containing phenolate groups (i.e., **L34**; Figure 25). This structure contains three labile protons, which through tautomerism provides an anionic N_2_O_2_ coordination core that accommodates a number of divalent (i.e., Cu^II^ and Ni^II^) and trivalent (i.e., Co^III^ and Mn^III^) transition metal cations, forming intensely colored tricyclic MN_2_O_2_ chelates endowed with redox properties (i.e., **111**–**114**; Figure 25). In the complexes containing divalent metals the ligand is dianionic, losing only the two phenolic protons, whereas in the complexes containing trivalent metals the ligand is triply deprotonated. The resulting geometries around the metal center are square-planar and distorted octahedral for the bivalent and trivalent complexes, respectively. All complexes were obtained by reacting the ligand with the corresponding divalent salt chlorides in DMF. However, Co^II^ and Mn^II^ derivatives underwent spontaneous oxidation in air, affording the trivalent complexes. In each case, the ligand **L34** behaved as a tetradentate ligand. Later on, only the complex containing manganese(III) (i.e., **114**) was exhaustively investigated both in vitro and in vivo for its biological activity profile by R. Farghadani et al. [68,69]. The in vitro studies entailed the assessment of their antiproliferative activity against two human mammary carcinoma cell lines (i.e., hormone-dependent MCF-7 and triple negative breast cancer MDA-MB-231 cells) and two human normal cell lines (i.e., 184B5 breast cells and WRL-68 hepatic cells). According to the results obtained by MTT method, **114** exerted significant (IC_50_ in the low micromolar range) reduction of the cell viability of MCF-7 and MDA-MB-231 cells with no effect against the two non-tumorigenic cell lines. Further cytotoxicity assays, such as trypan blue staining and lactate dehydrogenase, performed on the two tumorigenic cell lines, evidenced that **114** decreases the cell viability in a dose-dependent manner and induces cell death after cell membrane disruption, suggesting an apoptosis mechanism. The latter was confirmed by morphological studies (Hoechst 33342/PI and Annexin V/PI assays) which evidenced significant changes with nuclear condensation and externalization of phosphatidyl serine. Flow cytometry analysis showed that this β-diiminate Mn(III) complex induce cell cycle arrest at the G_0_/G_1_ phase. Using fluorescent probes, the authors were also able to demonstrate that **114** causes disruption of mitochondrial membrane potential and substantial ROS accumulation, highlighting the involvement of the mitochondrial intrinsic pathway in the mechanism of cell death [68]. Subsequent in vitro assessments showed that **114** induces apoptosis by increasing also caspase-3/7 activity and that it is capable to exert a synergistic antiproliferative effect in combination with doxorubicin (but not with tamoxifen). Furthermore, gene expression analysis revealed that **114** exerts its antiproliferative effect through the up- and down-regulation of p21 and cyclin D1, respectively, along with increased expression of the Bax/Bcl-2 ratio. The safety profile of this complex was evaluated in vivo in a murine model of acute oral toxicity. This test revealed that there was no toxicity or mortality on the treated animals compared to the control group [69].

### 3.4. Polypyridyl Complexes

In order to investigate photophysical and electrochemical properties of indole-containing metal complexes, Jason Shing-Yip Lau et al. synthetized luminescent cyclometalated iridium(III) polypyridine derivatives with general formula ([Ir(N^∧^C)_2_(N^∧^N)](PF_6_) (i.e., **115**–**117**; Figure 26) by reaction of [Ir_2_(N^∧^C)_4_Cl_2_] (HN^∧^C) Hppy, Hbzq, or Hpq with two equivalents of ligand bpy-ind or bpy-C6-ind (Figure 26) [67]. The complexes were characterized by means of IR and NMR spectroscopy. Moreover, the electrostatic interaction of the complexes with the indole-binding protein BSA was assessed using emission titrations. According to the obtained data, emission and lifetime extension of the complexes have enhanced upon binding to BSA, due to the increased hydrophobicity and rigidity of the local environments of the complexes. The cytotoxicity of these complexes against HeLa cells was investigated by MTT assay providing IC_50_ values in the low micromolar range (1.1–6.3 µM), much lower than cisplatin (30.7 µM) and related indole-free complexes evaluated in the same experimental conditions. Flow cytometry and laser-scanning confocal microscopy studies highlighted efficient uptake of the complexes **115a**–**117a** by HeLa cells with localization in the perinuclear region. In particular, **117a** displayed the highest emission intensity. The mechanism of internalization of **117a** was evaluated by comparative studies with Ir-BSA and Ir-TF (TF = holo-transferrin) luminescent bioconjugates, which exhibited the same intracellular localization. This outcome suggested that **117a** enters HeLa cells through an endocytosis mechanism. The same experiments performed in the presence of cytoskeletal (nocodazole and colchicine) and ATPase (potassium nitrate, sodium orthovanadate(V) and sodium molybdate(VI) dehydrate) inhibitors did not bring about significant changes in the cellular uptake and localization of **117a**, therefore excluding the involvement of these pathways in its internalization process. When performed in the presence of the oxidative phosphorylation inhibitor carbonyl cyanide 3-chlorophenylhydrazone instead, these experiments displayed a substantial decrease of cellular-uptake of **117a** suggesting that the process occurs via energy-requiring endocytosis (confirmed also by temperature-dependence experiments) [70].

### 3.5. N, S-Donor

In the context of the metal complexes bearing indole-containing N,S-donor ligands, the more extensive work has certainly been carried out by R. Karvembu’s research group. In the first research work, Jebiti Haribabu et al. explored the DNA-binding capacity of metal complexes by means of absorption/emission spectroscopy, viscosity measurements and EB competitive binding studies [71]. Using indole-based thiosemicarbazone ligands (**L35**–**L38**; Figure 27) as starting materials, a small set of Ni(II) complexes [Ni{C_10_H_9_N_2_NHCSNH(R)}_2_] where R = hydrogen (**118**), 4-methyl (**119**), 4-phenyl (**120**) and 4-cyclohexyl (**121**); Figure 27) was obtained and characterized by elemental analysis, UV–Vis, FT-IR, ^1^H and ^13^C NMR and mass spectroscopic techniques. The molecular structure of the ligands **L37** and **L38** and complexes **119**, **120** and **121** (square-planar geometry) was also confirmed through X-ray diffraction analysis (Figure 28). The investigation of their biological activity revealed the intercalative interaction of the complexes with CT-DNA assessed by EB method. Furthermore, based on the data obtained by UV–Vis, fluorescence and synchronous fluorescence spectroscopic methods, the compounds displayed strong BSA-interaction. All Ni(II) complexes displayed high antioxidant activity (especially complex **120**; IC_50_ = 50 µg/mL) assessed by DPPH method) and did not induce any hemolytic activity. Gel electrophoresis experiments demonstrated that these complexes exert DNA-cleavage activity without any external agent. They also showed moderate to remarkable anticancer activity against lung cancer (A549), human breast cancer (MCF7) and mouse embryonic fibroblasts (L929) cell lines. Complex **121** possessed the highest cytotoxicity against both tumor cell lines and the lowest toxicity in normal cells (IC_50_ > 750 µM). The authors, therefore, chose **121** to evaluate the mechanism of cell death using the Hoechst 33258 staining method, which highlighted that it occurs via apoptosis [71].

Pd^II^ complexes [PdCl(L)(PPh_3_)] (**122**–**126**) and [Pd(L)_2_] (**127** and **128**; Figure 29) were synthesized later on by the same authors using the same indole-3-carbaldehyde thiosemicarbazone ligands **L35**–**L38** plus the ethyl derivative **L39** (Figure 29) [72]. The structure of the new ligand **L39** and complexes **123**–**127** were determined by X-ray analysis, which revealed distorted square-planar geometries for complexes in which the thiosemicarbazone moiety acts as a bidentate monobasic (NS−) ligand and is coordinated to the Pd^II^ ion in such a way that it forms a five membered ring and the remaining sites are occupied by one chlorine and one triphenylphosphine (e.g., **123** and **127**; Figure 30). On the contrary, complex **127** adopted a square-planar geometry, forming two five-membered rings in which two indole-bound thiosemicarbazone ligands are coordinated to the Pd^II^ ion in a trans fashion. The complexes bound efficiently to CT-DNA (EB displacement assay) via intercalative binding mode and significantly cleaved the DNA (pUC19 and pBR322) with no presence of co-oxidant at pH 7.2 and temperature 37 °C. Their DNA-binding affinity followed the order **125** > **126** > **124** > **122** > **123**. The higher binding propensity of the complexes **125** and **126** might be due to the presence of the bulky cyclohexyl and phenyl group at the N-terminal position, respectively. The binding with the intended target was corroborated by docking studies, which highlighted effective interactions with B-DNA dodecamer (PDB ID: 1BNA) and human DNA-Topoisomerase I complex DNA-T and (PDB ID: 1SC7) for all complexes (which also bound efficiently to BSA). Additionally, the antiproliferative activity of the complexes against HepG-2, A549 and MCF7 cancer cells and one normal cell line (L929) was evaluated. All complexes exhibited acceptable cytotoxicity only against HepG-2 cells. In particular, complexes **125** and **126** containing the triphenylphosphine group showed the highest activity with IC_50_ values of 22.8 and 67.1 μM, respectively. Furthermore, complex **125** exhibited an activity almost equivalent to that of cisplatin. Noteworthy, the toxicity of all the complexes towards the normal cell line was lower than that found towards the tumor cell lines. DNA fragmentation and fluorescence staining studies highlighted considerable morphological changes in HepG-2 cells, suggesting that cell death occurred by apoptosis [72].

More recently, the research group of R. Karvembu carried out a study on a water-soluble binuclear organometallic Ru(II)-p-cymene complex ([Ru(η^6^-p-cymene)(η^2^-L)]_2_, (**129**; Figure 31) prepared from the reaction of the ligand **L37** [i.e., (E)-2-((1H-indol-3-yl)methylene)-N-phenylhydrazine-1-carbothioamide] with [RuCl_2_(p-cymene)]_2_, and its structure was analyzed by UV–Vis, FT-IR, NMR and mass spectroscopic analyses [73]. In addition, the structure of the binuclear complex was determined by X-ray crystallography (Figure 32). Based on the data obtained, a pseudo-octahedral geometry was hypothesized for the Ru(II) complex with the auxiliary ligand p-cymene and a (N,S) TSC chelating ligand located around each Ru(II) ion, which, in turn, is connected by two-bridged sulfur atoms of the TSC ligands. The antiproliferative assay (MTT method) performed for both the ligand its related complex **129** against A549-lung, MCF-7-breast, HeLa-cervical, HepG-2-liver, T24-urinary bladder and EA.hy926-endothelial cancer cells, and Vero-kidney epithelial normal cells highlighted significant activity for the complex against A549, HeLa and T24 cancer cells, with IC_50_ values lower than that of cisplatin (e.g., complex → IC_50_ = 7.70 μM vs. cisplatin → IC_50_ = 18.0 μM in A549), (complex → IC_50_ = 11.2 μM vs. cisplatin → IC_50_ = 22.4 μM in HeLa cells) and (complex → IC_50_ = 5.05 μM vs. cisplatin → IC_50_ = >50 μM in T24 cells). Moreover, the authors carried out in silico molecular docking studies, which suggested that the two compounds might be investigated as antiviral agents since they showed potential binding to the spike protein and main protease of SARS-CoV-2 [73].

Ru(II) complexes (i.e., **130**–**133**; Figure 33) were synthesized from O-R-1H-indole-2-carbothioate ligands **L44**–**L47** (which were obtained in turn from ligands **L40**–**L43**; Figure 33) and characterized using ^1^H and ^13^C NMR spectroscopy, and high-resolution ESI-MS. Moreover, ligand **L47**, complexes **131**, **132** and **133** were analyzed by single-crystal X-ray diffraction (Figure 34), which revealed that the complexes adopt pseudo-octahedral structures in which there is a η6-p-cymene ring, a N,S-chelated indole and a chloride ion to make 18-electron complexes with “piano-stool” geometry. All complexes were tested as antibiotic agents against *M. abscessus* NCTC 13031, *E. coli* ATCC 11775, I469 ESBL, J53 2138E, J53 2140E, *S. aureus* ATCC 29213, *A. baumannii* NCTC 12156, *S. enterica* serovar typhi and *M. tuberculosis* H37Rv. The antimicrobial assays showed that complex **132** was the most effective derivative as it inhibited nine out of the twelve organisms tested; to follow, the most active derivative was found to be complex **131**. This outcome might be ascribed to the steric hindrance of the R alkyl group on the indole ring which in turn may have an effect on the aquation rate and degree of diffusion across biological barriers. Additionally, complexes **130**, **131** and **132** exhibited moderate cytotoxicity against A2780 and A2780cisR cancer cell lines (assessed by MTT assay) and even lower activity (~2–3 folds) against normal prostate epithelial cells PNT2 [74].

Four Zn(II) complexes (i.e., **134**–**137**; Figure 35), composed of the Zn^2+^ ion coordinated to two previously disclosed indole-based thiosemicarbazone ligands (i.e., **L35**–**L36** and **L38**–**L39**; Figure 35), were designed and synthesized by N. Balakrishnan et al. [75]. They were characterized by spectroscopic techniques such as UV–Vis, FT-IR, ^1^H NMR, ^13^C NMR and MS. The structures of **134** and **136** were determined by X-ray diffraction methods, which showed that in both Zn^2+^, in a distorted tetrahedral environment, was coordinated to the azomethine N and thiocarbonyl S atoms. The indole-thiosemicarbazone unit acted as a bidentate ligand, with two of them coordinated with the Zn^2+^ ion (Figure 36). Moreover, the binding affinity of the complexes to DNA was investigated by UV–Vis spectroscopy and viscosity measurements, which indicated that the cyclohexyl derivative **137** had the strongest ability to bind DNA and that all complexes bound to DNA by an intercalation mechanism. These complexes also showed effective BSA-binding through a static quenching mechanism; complex **137**, however, bound to the protein more strongly than the other complexes. The DNA/BSA-binding trend of the complexes was also confirmed by docking studies. The cytotoxicity studies, evaluated by MTT method using two human cancer cell lines (A549 and MCF7), two human non-tumorigenic cell lines (MCF-10A and HEK-293) and one non-cancerous mouse fibroblasts (L929) cell line, revealed that complex **137** was the most effective derivative against A549 and MCF7 cells with IC_50_ = 37.9 and 60.3 µM, respectively. Its cytotoxicity was comparable to that of cisplatin and, more importantly, complex **136** and **137** showed remarkable tumor selectivity. The mechanism of cell death was investigated by Hoechst 33258 staining assay, which evidenced that **137** cause morphological changes in cancer cells typical of the apoptotic pathway [75].

## 4. Insights into Mechanisms of Action of Indole-Containing Metal Complexes

Based on what has been presented in the various sections of this article, it is clear that the indole ring plays an extraordinarily important role in the mechanisms of action of complexes that contain it in their structure, and that these are manifold. Its electron-rich planar aromatic structure not only forms a hydrophobic environment essential for interaction and stabilization with biological targets such as nucleotides and proteins, but it can also be an active component of reactions involving electron-transfer pathways expressed at the mitochondrial level. Of particular relevance in the antitumor field are the π–π stacking interactions that indole-containing metal complexes form by intercalating with DNA base pairs, resulting in disassembly and/or disruption of gene structure and blockage of replication. In addition, as we have seen, several enzymatic (e.g., Topoisomerase I) and receptor (e.g., kinase proteins) proteins can be inhibited or modulated by these types of complexes, again because of the indole component in the structure. In some cases, the indole fragment has been related to targeting of complex intracellular structures such as microtubules and inhibition of mitosis. In the antimicrobial context, the mechanistic role played by the indole element is mostly related to its chelating properties and the enhanced lipophilicity of the resulting complexes with transition metals. In the chelation process, the orbital of the ligand tends to overlap with the positive charge of the metal ion, reducing its polarity and increasing the π–electron delocalization of the whole chelate ring system. This results in increased lipophilicity of complexes and their increased diffusion across lipid membranes. Once inside microorganisms, some metals ions can inactivate their enzymes or generate lethal ROS. The mechanism of killing microbes can otherwise be explained on the basis of the chelating effect of indole-containing ligands, which can bind metal cofactors of metalloproteins by disturbing the homeostasis of microbial cells.

## 5. Conclusions

Indole-containing metal complexes have been shown to successfully improve selectivity and therapeutic potential of parent compounds lacking the metal component or the indole scaffold. Their potential use in medicine has been demonstrated to be considerable, including anti-cancer, anti-tubercular, antiviral, and antibacterial applications. Coordination with various transition metals, such us Pd(II), Zn(II), Mn(II), Sn(IV), Co(II), Fe(II), Ni(II), Cu(I), Cu(II), Ir(II), Rh(II), Ag(I) and Ru(I) resulted in a considerable increase in the efficacy, as highlighted by several in vitro studies on different types of biological experiments (cell-based and enzymatic). In many cases, their IC_50_ values were found to be lower than those of standard reference drugs. Both the presence of the indole scaffold and transition metal correlate with increased activity of the resulting complexes as evidenced also by computational studies. Characterization studies highlighted the impact of the presence of these two chemical features on the overall geometry of the complexes which in turn plays a key role in the binding with biological targets. Furthermore, coordination with transition metals often determines a significant increase in the solubility of the complexes, thus leading to a better ADME profile and a consequent superior therapeutic applicability. In vivo studies on this type of complexes are still at a preclinical level but constitute an area of active research (especially in the antitumor field), since in some cases they have demonstrated to possess an excellent safety and tolerability profile in murine models.

The development of indole-containing metal complexes and their enhanced activity may constitute a concrete opportunity to support the standard of care on various therapeutic approaches.

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
