# Peer review of "Indole-Containing Metal Complexes and Their Medicinal Applications"

_molecules, 2024, doi:10.3390/molecules29020484_

Round 1

Reviewer 1 Report

Comments and Suggestions for Authors

The work is a routine one of marginal interest, falling short of the criteria for publication in such an esteemed journal. The authors are advised to address the following issues before considering resubmission to an alternative journal.

-The main purpose of this review appears ambiguous, with uncertainty regarding whether it is focused on synthesis or biological applications. It seems that the authors were unable to effectively convey the actual purpose of this study. Clarifying this aspect would significantly enhance the manuscript's overall coherence and impact.

-The authors are encouraged to incorporate a section detailing the mechanism of action of the complexes in their manuscript. This addition will contribute to a more comprehensive understanding of the study's findings and provide valuable insights into the functioning of the investigated complexes.

Comments on the Quality of English Language

The manuscript is well-written with clear and effective language.

Author Response

Reviewer #1

Comments and Suggestions for Authors

The work is a routine one of marginal interest, falling short of the criteria for publication in such an esteemed journal. The authors are advised to address the following issues before considering resubmission to an alternative journal.

- We thank the Reviewer for his/her commitment and appreciate the critical general comment (which is always to be considered positive because it helps us to improve and delve deeper into the problems). We followed his/her advises by solving the proposed issues and we hope to have managed to arouse his/her interest and the level of the article which we consider appropriate for the Special Issue we were invited, i.e. “The Design and Synthesis of Indole Derivatives and their Metal Complexes”. You will see all revisions highlighted in yellow color.  

-The main purpose of this review appears ambiguous, with uncertainty regarding whether it is focused on synthesis or biological applications. It seems that the authors were unable to effectively convey the actual purpose of this study. Clarifying this aspect would significantly enhance the manuscript's overall coherence and impact.

- Yes, we agree with this observation. The purpose of the review article is to highlight the biological activity profile of indole-containing metal complexes with a perspective for future medical applications. Therefore, we revised the manuscript accordingly, trying to avoid the misleading term “synthesis” in the abstract and premise sentence and replacing it with “development”. Syntheses of course are still briefly reported for information purposes for the reader but are not of primary importance as other reviews deal extensively with this topic. Based on the above, we have delved into the biological section of each article we have cited.

-The authors are encouraged to incorporate a section detailing the mechanism of action of the complexes in their manuscript. This addition will contribute to a more comprehensive understanding of the study's findings and provide valuable insights into the functioning of the investigated complexes.

-Thanks for this precious suggestion! Since the mechanisms of action involved are often multiple for these complexes (in some cases even unclear or simply inferred indirectly), we thought it best to analyze the cited articles in more detail and report mechanistic details for each regarding biological activity. We have therefore expanded (where possible) this part article by article. In any case, so as not to leave your request unanswered, we have also added a paragraph before the conclusions entitled "Insights into mechanisms of action of indole-containing metal complexes" that summarizes what we found. We hope this is convincing for you.

Reviewer 2 Report

Comments and Suggestions for Authors

Kazemi et al. did a review of metal complexes with ligands containing an indole moiety. I am critical about this manuscript. In my opinion, it does not really create added value and it lacks a clear theme. However, if so desired, it may be published in the special issue entitled "The Design and Synthesis of Indole Derivatives and Their Metal Complexes" after minor revision.

"Indole-based metal complexes" in the title might suggest that only metal complexes with an indole moiety as coordinating group are comprised. This is, however, not the case, beause also ligands having an indole moiety in the periphery are covered.

In the context of medicinal chemistry, I do not find the classification by the type of ligand (e.g. monodentate and so on) particularly useful. One would rather expect an arrangement of the compounds by the field of (potential) applications.

Line 14: "indole nucleus" sounds perhaps a bit strange. Maybe "indole group", "indole moiety" or "indole scaffold" or so.

Line 23: Hückel's rule is theoretically justified only for monocyclic compounds. Therefore, it is probably inappropriate to state that "According to Huckel's rule, indole is aromatic in nature".

Line 81: The "As a matter of fact," is unnecessary.

Scheme 1 is a bit strange. In formula A, the indole nitrogen atom has five bonds, which violates the octet rule. Is that intentional? Sure that formula C is best described as pi complex? Maybe it is rather a sigma complex like formula A.

Line 121: I am not sure if the complex formation is best described as "condensation".

Figure 32: In compounds 126-129, the indole nitrogen atom in the chemical diagram should probably not carry a hydrogen atom. The style of the structure pictures of compounds L51 and 129 looks different from the other structure pictures, because they were obviously drawn with another program (probably Mercury). It would look neater if the style of the structure representations were the same throughout the article.

I wonder if the conclusion "Indole-based metal complexes have showed to be able to successfully enhance selectivity and therapeutic efficiency." is really justified, because in vivo data for these metal complex appear to be scarce, not to speak of clinical data.

Author Response

Reviewer #2

Comments and Suggestions for Authors

Kazemi et al. did a review of metal complexes with ligands containing an indole moiety. I am critical about this manuscript. In my opinion, it does not really create added value and it lacks a clear theme. However, if so desired, it may be published in the special issue entitled "The Design and Synthesis of Indole Derivatives and Their Metal Complexes" after minor revision.

We thank the Reviewer for his/her commitment in revising the article and for the critical comment. We regret that the purpose of the article wasn’t unclear. Regardless, we have taken his/her comments into account as a  compelling input for improving the quality of the article, and we believe that this helped us very much. In order to make the theme of the review more clear, the aim of which is to describe the different indole-containing metal complexes and their potential use in medicine, we decided to delve into the biological experiments section of each article and discuss them more extensively. We also included, based on another Reviewer’s request, a small section dealing with the mechanistic studies of this type of metal complexes. You will see all revisions highlighted in yellow color. Again, we thank you for this critical boost and we hope the article in now convincing for you. 

"Indole-based metal complexes" in the title might suggest that only metal complexes with an indole moiety as coordinating group are comprised. This is, however, not the case, because also ligands having an indole moiety in the periphery are covered.

Yes, we agree with this observation. Accordingly, we changed the title to "Indole-containing metal complexes and their medicinal applications". In the main text we left the term “indole-based” only in the few cases we are talking about ligands.

In the context of medicinal chemistry, I do not find the classification by the type of ligand (e.g. monodentate and so on) particularly useful. One would rather expect an arrangement of the compounds by the field of (potential) applications.

This observation is somehow understandable but a classification based on the potential applications is even more complicate because in many cases the same metal complexes undergo different biological assessments (antimicrobial, antiviral, anticancer, and so forth …). Then, we believe that the classification by the type of ligand is still useful and allows to highlight the different modes of metal-ligand coordination clearly and schematically. However, as stated above, we added an in-depth analysis of the biological tests. So, we believe that it is fine this way.

Line 14: "indole nucleus" sounds perhaps a bit strange. Maybe "indole group", "indole moiety" or "indole scaffold" or so.

Corrected accordingly.

Line 23: Hückel's rule is theoretically justified only for monocyclic compounds. Therefore, it is probably inappropriate to state that "According to Huckel's rule, indole is aromatic in nature".

Actually, indole has a planar, cyclic structure and 10 pi electrons, which means it satisfies the 4n+2 rule. Not only monocycles satisfy Hückel's rule (e.g. Naphthalene). We therefore believe that the sentence is correct. Besides, many other articles state it this way.

Line 81: The "As a matter of fact," is unnecessary.

Removed accordingly.

Scheme 1 is a bit strange. In formula A, the indole nitrogen atom has five bonds, which violates the octet rule. Is that intentional? Sure that formula C is best described as pi complex? Maybe it is rather a sigma complex like formula A.

We apologize for the mistake in formula A of Scheme 1. It was revised accordingly. Regarding the formula C, Yamamoto et al, have demonstrated a σ–π continuum for the indole-metal interaction, while the dominant coordination mode is the Wheland-intermediate type σ-mode considering the strong electron-donating ability of indole. Nonetheless, there was still an error in formula C. So, thanks for paying full attention to this scheme.

Line 121: I am not sure if the complex formation is best described as "condensation".

Yes, you’re right. The word “condensation” was changed with “reaction”.

Figure 32: In compounds 126-129, the indole nitrogen atom in the chemical diagram should probably not carry a hydrogen atom. The style of the structure pictures of compounds L51 and 129 looks different from the other structure pictures, because they were obviously drawn with another program (probably Mercury). It would look neater if the style of the structure representations were the same throughout the article.

Yes, once again you’re right. We thank the reviewer for his/her report. The figure has been modified and standardized with the others.

I wonder if the conclusion "Indole-based metal complexes have showed to be able to successfully enhance selectivity and therapeutic efficiency." is really justified, because in vivo data for these metal complex appear to be scarce, not to speak of clinical data.

We believe that the indole-containing metal complexes have demonstrated excellent results in vitro, often showing better selectivity towards tumor cells compared to healthy ones. This demonstrates that they are certainly worthy of further study. No clinical trials have been found at this time and there are very few reports on in vivo testing. Those we found have now been inserted and discussed in the revised version manuscript. However, as we stated in the reworked final paragraph “In vivo studies on this type of complexes are still at a preclinical level but constitute an area of active research (especially in the antitumor field) since in some cases they have demonstrated to possess excellent safety and tolerability profile in murine models”.

Round 2

Reviewer 1 Report

Comments and Suggestions for Authors

After careful consideration, it appears that the manuscript has not demonstrated significant improvements.  Continuing with my previous assessment I, recommend rejecting the manuscript once again. 

Comments on the Quality of English Language

The manuscript was written in good English language.